# Characteristics of Free Amino Acids (the Quality Chemical Components of Tea) under Spatial Heterogeneity of Different Nitrogen Forms in Tea (*Camellia sinensis*) Plants

**DOI:** 10.3390/molecules24030415

**Published:** 2019-01-24

**Authors:** Li Ruan, Kang Wei, Liyuan Wang, Hao Cheng, Liyun Wu, Hailin Li

**Affiliations:** 1National Center for Tea Improvement, Tea Research Institute, Chinese Academy of Agricultural Sciences, Hangzhou 310008, China; ruanli@tricaas.com (L.R.); weikang@tricaas.com (K.W.); wuly@tricaas.com (L.W.); lihailin@tricaas.com (H.L.); 2State Key Laboratory of Soil and Sustainable Agriculture, Institute of Soil Science, Chinese Academy of Sciences, Nanjing 210008, China

**Keywords:** nitrogen forms, heterogeneity, free amino acids, tea quality, root development

## Abstract

Nitrogen (N) forms are closely related to tea quality, however, little is known about the characteristics of quality chemical components in tea under the spatial heterogeneity of different N forms. In this study, a split-root system, high performance liquid chromatography (HPLC), and root analysis system (WinRHIZO) were used to investigate free amino acids (FAAs) and root length of tea plants under the spatial heterogeneity of different N forms. Uniform. (U.) ammonium (NH_4_^+^) (both compartments had NH_4_^+^), U. nitrate (NO_3_^−^) (both compartments had NO_3_^−^), Split. (Sp.) NH_4_^+^ (one of the compartments had NH_4_^+^), and Sp. NO_3_^−^ (the other compartment had NO_3_^−^) were performed. The ranking of total FAAs in leaves were as follows: U. NH_4_^+^ > Sp. NH_4_^+^/Sp. NO_3_^−^ > U. NO_3_^−^. The FAA characteristics of Sp. NH_4_^+^/Sp. NO_3_^−^ were more similar to those of U. NO_3_^−^. The contents of the important FAAs (aspartic acid, glutamic acid, and theanine) that determine the quality of tea, increased significantly in U. NH_4_^+^. The total root length in U. NH_4_^+^ was higher than that in the other treatments. More serious root browning was found in U. NO_3_^−^. In conclusion, NH_4_^+^ improved the accumulations of FAAs in tea leaves, which might be attributed to the root development.

## 1. Introduction

As a leaf crop, tea plants consume large amounts of nitrogen (N). For tea plants, N uptake accounts for 4.5% of the dry weight. Thus, N gives great contributions to the improvement of tea quality and yield [1]. Different N sources in the environment will affect the absorption of N nutrients in tea plants, and then affect the quality of tea [2]. Ammonium (NH_4_^+^) and nitrate (NO_3_^−^) are two main inorganic N forms sources in soil for plants. Due to many factors, including root activity, soil moisture, microbial activity, and so on, the conversion of NH_4_^+^ and NO_3_^−^ occurs frequently in soils [3]. Therefore, the root system can absorb either or both of these N forms [4]. Although both NH_4_^+^ and NO_3_^−^ can be used by plants, NH_4_^+^ and NO_3_^−^ have different absorption and assimilation mechanisms. This leads to different preferences of different plants [5]. In studies on tea plants, the responses to NH_4_^+^ and NO_3_^−^-supply have mainly focused on the absorption rates of NH_4_^+^ and NO_3_^−^, N use efficiency and N-related gene expressions [1,6,7,8,9]. Moreover, all of the previous studies were based on the condition of homogeneous N supply. Little is known about the changes of the biochemical components in tea under the spatial heterogeneity of N forms. For tea plants, amino acids are the most important chemical components that affect tea quality. For instance, glutamic acid has the umami taste and can improve the taste of tea infusions [10,11,12]. In addition, there was a significant positive correlation between tea quality and theanine content [13]. There are two main kinds of amino acids in tea plants: one is the amino acid which composes protein (relatively stable and rarely changes with the external environment); the other is the free amino acid (FAA) (the main form of N storage in tea plants that always change with the environment) [14]. In tea plants, a large proportion of amino acids are synthesized in the roots and are then transported to leaves. In other words, tea roots contribute greatly to amino acid synthesis [8]. Therefore, we proposed that tea roots can rapidly perceive changes in the external N form, and the FAA characteristics of leaves would change rapidly. In order to mimic a heterogeneous N form environment, we used the split-root system with different forms of N supply. In this research, FAAs and root length were analyzed by high performance liquid chromatography (HPLC) and root analysis system (WinRHIZO), respectively. The results of this study may contribute to a better understanding of the quality chemical component changes of tea plants under the spatial heterogeneity of different N forms.

## 2. Results and Discussion

### 2.1. Nitrogen (N) Content and Diversity of Free Amino Acids (FAAs) in Leaves

In order to study the response of tea plants to different N forms, we used a split-root system in the current research. In this split-root system, different N forms were provided in the two separate spaces. Three treatments were designed, as shown in Figure 1a. The N content in leaves is shown in Figure 1b. The ranking of the N content in the leaves was as follows: U. NH_4_^+^ > Sp. NH_4_^+^/Sp. NO_3_^−^ > U. NO_3_^−^. There were no significant differences in leaf N content among the three treatments after three days of treatment with different N forms. As a leaf crop, tea plants consume relatively high amounts of N; therefore, this nutrient plays a key role in improving tea yield and quality [1]. The leaf N content is associated with the quality of tea [2]. After short-term treatment with different N forms, differences in the leaf N content started to appear among the treatments. This suggested that tea plants had different utilization efficiency for different N forms. FAAs, which are sensitive to the environment, are the main components that determine the quality of tea [14]. The diversity of the FAAs in the leaves is shown in Figure 1c. Principal component analysis of FAAs diversity could clearly separate the three treatments, suggesting that the contents of FAAs changed with different N form treatments. In the direction of PCA1, the distance between the Sp. NH_4_^+^/Sp. NO_3_^−^ treatment and the U. NO_3_^−^ treatment was closer, indicating that the FAA characteristics of treatment Sp. NH_4_^+^/Sp. NO_3_^−^ were more similar to those of the treatment U. NO_3_^−^, rather than the treatment of U. NH_4_^+^. This suggested that there was a signal exchange between the roots on both sides of the split-root system. The supply of NO_3_^−^ on one side had a great influence on the FAA assimilation on the other side. According to the vector length, aspartic acid, glutamic acid, theanine, and total amino acids were major contributors to the diversity of the FAAs in leaves, indicating that the differences in the FAAs were mainly caused by changes in aspartic acid, glutamic acid, theanine, and total amino acids. After short-term treatment with different N forms, aspartic acid, glutamic acid, theanine, and total amino acid changed first. Aspartic acid, glutamic acid, and theanine were more sensitive than other FAAs to the changes of external N forms.

### 2.2. Free Amino Acid (FAA) Content Characteristics in Leaves

In this study, 17 kinds of FAAs were detected in the tea plant leaves (Figure 2a). These 17 kinds of FAAs included aspartic acid, serine, glutamic acid, glycine, histidine, arginine, threonine, alanine, proline, theanine, cysteine, tyrosine, valine, methionine, lysine, isoleucine, and leucine. The contents of FAAs varied from 0.02 to 0.38 mg g^−1^ DW. Among these FAAs, glutamic acid, theanine, and aspartic acid predominated (Figure 2a). Among the three treatments, U. NH_4_^+^ had the highest contents of total and major FAAs (Figure 2b). The ranking of total FAAs in the leaves was as follows: U. NH_4_^+^ > Sp. NH_4_^+^/Sp. NO_3_^−^ > U. NO_3_^−^. There were significant differences in the total FAA of leaves among the three treatments. FAAs are closely related to the taste and aroma of the tea, which determines the quality of the tea [15]. Glutamic acid and aspartic acid, which have the umami taste, can improve the taste of tea infusions [10,11,12]. Theanine can reduce the bitter and astringent taste of tea infusions and improve the sweet and umami taste [16,17,18]. There is a significant positive correlation between tea quality and the theanine content [19,20]. After short-term treatment with U. NH_4_^+^, the contents of the three important amino acids (aspartic acid, glutamic acid, and theanine) that determine the quality of the tea, increased significantly. This suggested that tea plants had a preference for NH_4_^+^, which was consistent with many previous studies [1,6,21]. NH_4_^+^ absorbed by the tea roots is rapidly converted into amino acids in the roots. This process requires less material and energy than the assimilation and transport processes of other amino acid [22,23]. Unlike other plants, tea plants can convert excess glutamic acid into theanine, which is an amino acid unique to tea plants. On the other hand, NH_4_^+^ converts more easily to theanine than NO_3_^−^ [24]. These processes eliminated NH_4_^+^ toxicity, and made tea plants have a preference for NH_4_^+^ in this study.

### 2.3. Root Characteristics under Split-Root System with Different N Forms

After 30 days of treatment with different nitrogen forms using the split-root system, root development showed significant differences (Figure 3a). Many absorbing roots grew on both sides of the U. NH_4_^+^ treatment. A few absorbing roots grew on Sp. NH_4_^+^ side. However, hardly any absorbing roots grew on Sp. NO_3_^−^ and U. NO_3_^−^ sides (Figure 3a). The total root length in U. NH_4_^+^ was 2.67, 1.50, and 1.33 times as high as of that in U. NO_3_^−^, Sp. NO_3_^-^, and Sp. NH_4_^+^, respectively (Figure 3b). The total root length of the U. NH_4_^+^ treatment was significantly higher than that of the other treatments. There was no significant difference between Sp. NO_3_^−^ and Sp. NH_4_^+^ treatments. In a previous study, NH_4_^+^ promoted the lateral root growth of *Arabidopsis* in an ammonium transporter 1;3-dependent manner [25]. Moreover, NH_4_^+^ improved the expressions of the genes related to N absorption, metabolism, and cell growth in tomato under the localized supply of NH_4_^+^ [26]. In the current study, root development was improved by U. NH_4_^+^ treatment, which was consistent with the above studies [25,26]. As the most active part of the tea root system, absorbing roots contributed most to the absorption of water and most nutrients [27]. The U. NH_4_^+^ treatment may have improved the growth of absorbing roots, which helped increase the absorption of water and nutrients in tea plants. Due to the interference of Sp. NO_3_^−^, only a few absorbing roots grew on the Sp. NH_4_^+^ side. This suggested that there was a signal exchange between the roots on both sides of the split-root system. The supply of NO_3_^−^ on one side had a strong influence on the growth of the root system on the other side. In addition to the poor root growth, root browning was more serious in the U. NO_3_^−^ treatment. When NO_3_^−^ was supplied as a single N source, rhizosphere pH would be alkalized. Under high pH conditions, rice roots showed a browning phenomenon due to a large amount of iron and manganese oxides deposited on the root surface. This root browning would inhibit rice growth, which might be one of the reasons why rice had a preference for NH_4_^+^ [19]. Additionally, in this study, tea roots showed a browning phenomenon in the U. NO_3_^−^ treatment, which might have resulted in the limited nutrient uptake by the tea plants. An important cause of root browning is weak root respiration. NH_4_^+^ has been reported to improve root respiration, which can provide energy for the synthesis and transport of free amino acids in tea plants [28]. In addition, improved root respiration could reduce root browning, which might be another important reason why NH_4_^+^ promoted the root development of tea plants.

## 3. Materials and Methods

### 3.1. Plant Materials and Split-Root System

*Camellia sinensis*, variation Wuniuzao, was used in this study. Annual cutting seedlings of Wuniuzao were transplanted into a split-root system. In this split-root system, different N forms were provided in the two separate spaces. At the beginning of the split-root experiment, we selected tea plant seedlings with uniform root and shoot growth. Then, the tea roots were planted uniformly in two separated physical spaces uniformly. Three combinations of different N forms were provided as follows [29]: (1) a homogeneous NH_4_^+^ condition (U. NH_4_^+^: both compartments had NH_4_^+^), (2) a homogeneous NO_3_^−^ condition (U. NO_3_^−^: both compartments had NO_3_^−^), and (3) a heterogeneous split condition (Sp. NH_4_^+^/Sp. NO_3_^−^: one compartment had NH_4_^+^, and the other had NO_3_^−^). In the U. NH_4_^+^ and Sp. NH_4_^+^ treatments, the nutrient solutions contained the following macronutrients (mmol L^−1^): (NH_4_)_2_SO_4_ (1), KH_2_PO_4_ (0.07), K_2_SO_4_ (0.3), MgSO_4_·7H_2_O (1), CaCl_2_·2H_2_O (0.53), and Al_2_(SO_4_)_3_·18H_2_O (0.035) and micronutrients (μmol L^−1^) H_3_BO_4_ (7), MnSO_4_·H_2_O (1), ZnSO_4_·7H_2_O (0.67), CuSO_4_·5H_2_O (0.13), (NH_4_)_6_Mo_7_O_24_·4H_2_O (0.047), and EDTA-Fe (4.2). In the U. NO_3_^−^ and Sp. NO_3_^−^ treatments, the nutrient solutions contained the following macronutrients (mmol L^−1^): Mg(NO_3_)_2_ (1), KH_2_PO_4_ (0.07), K_2_SO_4_ (0.3), CaCl_2_·2H_2_O (0.53), and Al_2_(SO_4_)_3_·18H_2_O (0.035) and the same micronutrients. The pH of the three treatments was 5.0. Continuous aeration of the nutrient solution was carried out by pumps. The nutrient solution was replaced every three days. After three days of the above treatments, the tea plants were harvested for the determinations of amino acids and N contents in the leaves. After 30 days of the above treatments, the tea roots of each compartment were harvested to measure the root length. Three repeats (each repetition consisted of 20 plants) per group were used for this study.

### 3.2. Determinations of N and Free Amino Acid (FAA) Contents

After freeze drying and grinding, the N contents of the leaf samples were measured by a CN element automatic analyzer (Vario Max CN Analyzer, Elementar Analysensysteme GmbH, Frankfurt, Germany). To determine the FAA content, 0.2 g of dried and ground leaves were extracted by 10 mL pure water at a constant temperature of 100 °C for 30 min. The samples were shaken once for 10 min. Then, samples were centrifuged for 10 min at 3500 r min^−1^. The supernatant was taken and filtered through 0.22 μm the membrane filter (Membrane Solutions, Kent, WA, USA) of water-phase for use. The derivatization of free amino acids was performed using the AccQ-Fluor Reagent Kit (Waters, Milford, MA, USA) according to the manufacturer’s specifications. Briefly, 10 μL of standard amino acid mix solution and tea extracts were mixed with 70 μL of AccQ Tag borate buffer; then, 20 μL of AccQ Tag reagent, previously dissolved in 1.0 mL of AccQ Tag reagent diluents, was added. The reaction was allowed to proceed for 10 min at 55 °C. Separation was performed on the HPLC system (Waters, Milford, MA, USA) equipped with a Waters AccQ Tag reversed-phase HPLC column (150 mm × 3.9 mm, 4 μm). The mobile phase A consisted of AccQ Tag Eluent A Concentrate in deionised water (1:10 *v*/*v*). The mobile phase B was acetonitrile and the mobile phase C was ultrapure water. The running time was 45 min. Gradient elution was described in Appendix A. Sample injection volume was 5 μL. Flow rate was 1.0 mL min^−1^. Column temperature was set at 37 °C. Amino acids were detected at 248 nm, and identified by comparison of retention times and spectra of standard solutions of amino acids kit and l-theanine [30,31]. For quality control, we inserted the standard samples before, during, and after the sample testing. The relative standard deviations (RSDs) of the retention time and peak area of each amino acid were shown in Appendix A. The RSDs of the retention time and peak areas varied from 0.01% to 0.20%, 0.78% to 4.74%, respectively. These suggested that our test conditions were stable. The recoveries of FAAs varied from 84.6% to 102.3%.

### 3.3. Determination of Root Length

At the end of the experiment, the roots on each sides of the split root system were collected separately. The collected root samples were immediately stored in 25% alcohol at a temperature of 4 °C. An optical scanner (Epson, Nagano, Japan) was used to scan the root samples. A root analysis system (WinRHIZO) was used to analyze the total root length.

## 4. Conclusions

Under the spatial heterogeneity of N forms, the characteristics of free amino acids (FAAs) in tea were determined. The findings showed that the characteristics of the FAAs differed under the different N forms. Aspartic acid, glutamic acid, and theanine were more sensitive than other FAAs to the changes in external N forms. The U. NH_4_^+^ treatment improved the accumulations of total and major free amino acids in tea leaves. Due to the great influence of Sp. NO_3_^−^ on the other side of the compartment, the FAA characteristics of the Sp. NH_4_^+^/Sp. NO_3_^-^ treatment were more similar to those of the U. NO_3_^−^ treatment, rather than the treatment of U. NH_4_^+^. NH_4_^+^ promoted root growth and reduced root browning, which might have contributed greatly to the accumulations of free amino acids in tea leaves. Our findings will help us better understand of the quality chemical component changes of tea plants under the spatial heterogeneity of different N forms.

## Figures and Tables

**Figure 1 molecules-24-00415-f001:**
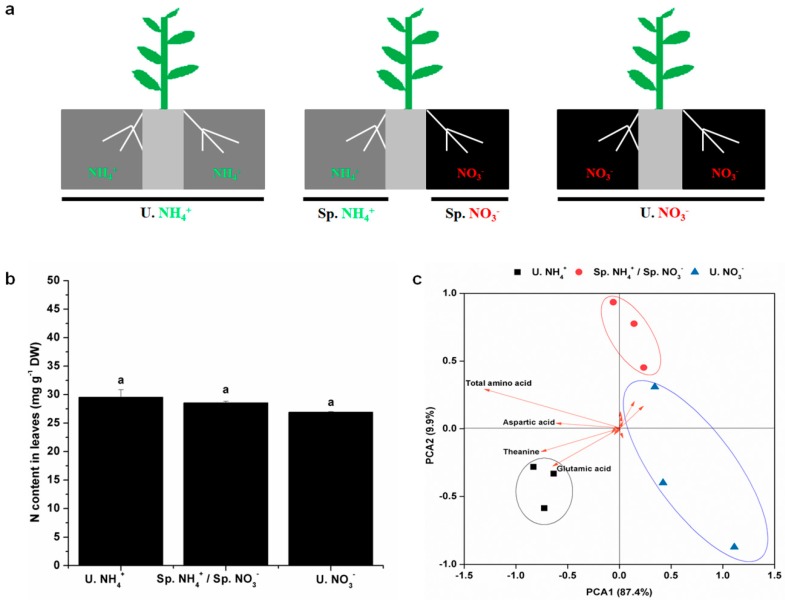
Nitrogen (N) content and diversity of free amino acids (FAAs) in leaves under the split-root system. (**a**) Diagram showing the split-root system used to simulate the spatial heterogeneity of different N forms. Two separated physical spaces where different N forms could be applied, were created in this split-root system: U. ammonium (NH_4_^+^) (both compartments had NH_4_^+^), U. nitrate (NO_3_^−^) (both compartments had NO_3_^−^), Sp. NH_4_^+^ (one of the compartments had NH_4_^+^), and Sp. NO_3_^−^ (the other compartment had NO_3_^−^); (**b**) leaf N content; (**c**) diversity of FAAs in the leaves. The mean ± SE (*n* = 3) are shown in the data. The different letters show that means are significantly different at *p* < 0.05. The different conditions to compare the analytical results obtained are shown in the Materials and Methods section.

**Figure 2 molecules-24-00415-f002:**
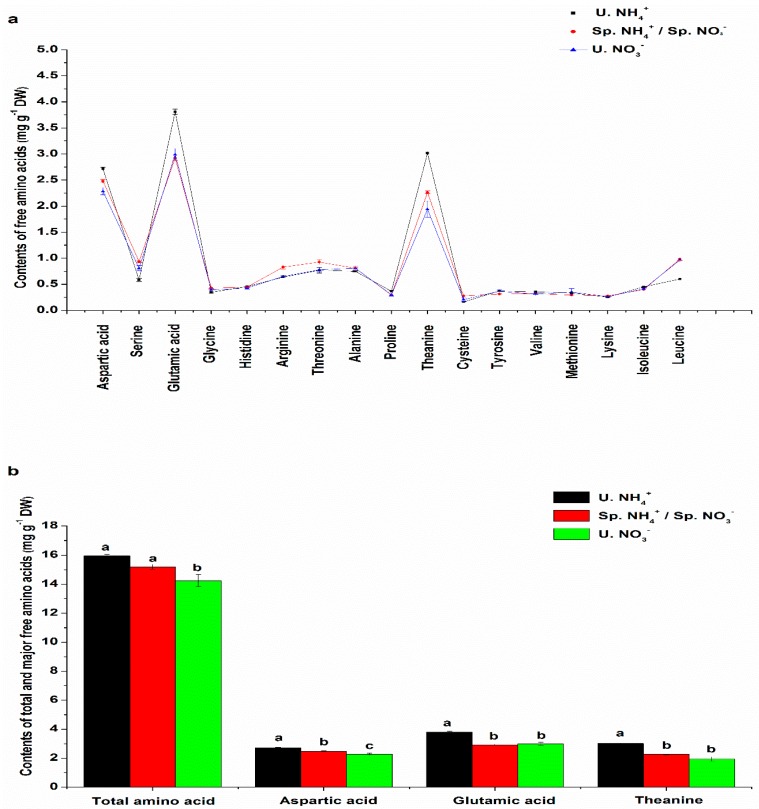
Free amino acid (FAA) content in the leaves. (**a**) The contents of all FAAs in the leaves; (**b**) the contents of total and major FAAs in the leaves. The mean ± SE (*n* = 3) are shown in the data. The different letters show that means are significantly different at *p* < 0.05. The different conditions to compare the analytical results obtained are shown in the Materials and Methods section.

**Figure 3 molecules-24-00415-f003:**
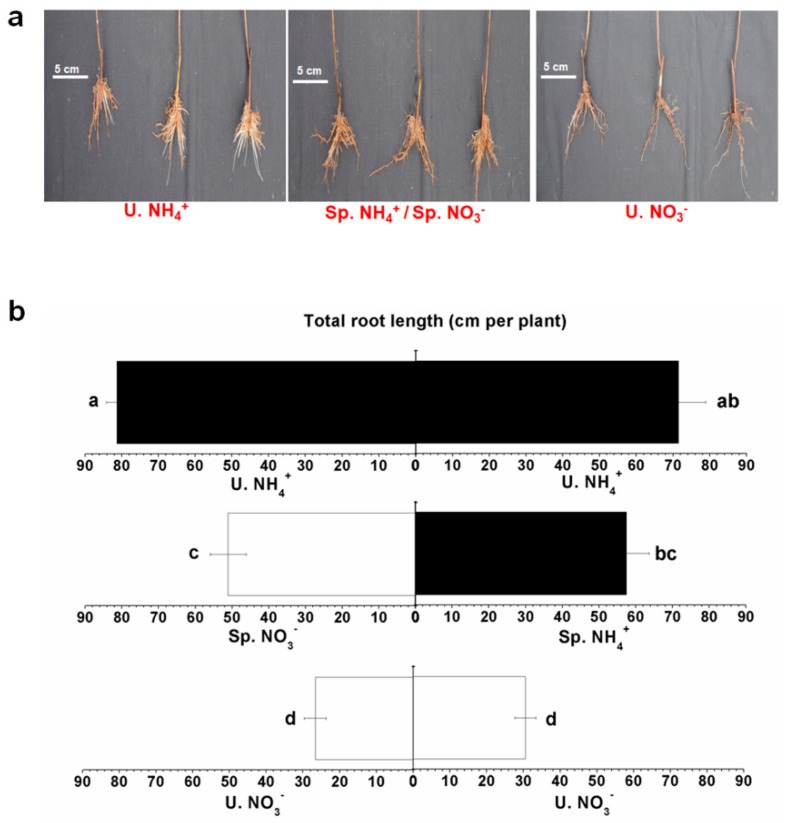
Root developments under the spatial heterogeneity of different N forms. (**a**) The root phenotypes of different treatments and (**b**) the total root length of different treatments. The mean ± SE (*n* = 3) are shown in the data. The different letters show that means are significantly different at *p* < 0.05. The different conditions to compare the analytical results obtained are shown in the Materials and Methods section.

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
