# Peer review of "Characteristics of Free Amino Acids (the Quality Chemical Components of Tea) under Spatial Heterogeneity of Different Nitrogen Forms in Tea (Camellia sinensis) Plants"

_molecules, 2019, doi:10.3390/molecules24030415_

Round 1

Reviewer 1 Report

The manuscript aims to study the differences in the free amino acid profile of tea leaves, when various forms of N are present in the environment. The topic is of great interest since the amino acid fraction contributes to the nutritional value and to the sensory properties of tea. 

Nevertheless, the analytical aspects regarding the determination of amino acids should be better described.

- SPE purification and derivatization conditions are not clear. For example, what type of column did the authors use?

- Is the analytical procedure used for the determnation of FAA validated? Can the authors provide LoD and LoQ? Which are the levels of precision?

-The authors state that "17 types of FAA" were detected. Why only results regarding aspartic acid, glutammic acid and theanine are presented?

-Can the authors provide a chromatogram of the FAA?

- How many plant samples per group were used for the study?

Author Response

Dear Editor and reviewer:

Thank you very much for giving us the opportunity to revise our manuscript.

We have studied reviewer’s comments and suggestions carefully and have made revision accordingly which are marked in red in the manuscript. We have also replied point by point to reviewer’s comments in the following letter which are marked in blue. Attached please find the revised version, which we would like to submit for your kind consideration. The English language of our manuscript has been editing by MDPI (english-7369).

We would like to express our great appreciation to you and reviewers for comments on our manuscript. Looking forward to hearing from you.

With best regards!

Yours sincerely,

Liyuan Wang, Hao Cheng

Reviewer 1

The manuscript aims to study the differences in the free amino acid profile of tea leaves, when various forms of N are present in the environment. The topic is of great interest since the amino acid fraction contributes to the nutritional value and to the sensory properties of tea. Nevertheless, the analytical aspects regarding the determination of amino acids should be better described.

1.     SPE purification and derivatization conditions are not clear. For example, what type of column did the authors use?

Response:

Thank you for your advice. Our original description is ambiguous. SPE purification and derivatization conditions have been added in the Materials and Methods section as follows (lines 182-190):

0.2 g of dried and ground leaves were extracted by 10 mL pure water at a constant temperature of 100 °C for 30 min. The samples were shaken once in 10 min. Then samples were centrifuged for 10 min at 3500 r min-1. The supernatant was taken and filtered through 0.22 um the membrane filter of water-phase for use. The derivatization of free amino acids was performed using the AccQ-Fluor Reagent Kit according to the manufacturer’s specifications. Briefly, 10 μL of standard amino acid mix solution, and tea extracts were mixed with 70 μL of AccQ Tag borate buffer; then 20 μL of AccQ Tag reagent previously dissolved in 1.0 ml of AccQ Tag reagent diluents were added. The reaction was allowed to proceed for 10 min at 55 °C (Joshi et al., 2015; Yu et al., 2014).

The type of column we used has been added in the Materials and Methods section as follows (lines 190-196):

Separation was performed on the HPLC system equipped with a Waters AccQ Tag reversed-phase HPLC column (150 mm×3.9 mm, 4 μm). The mobile phase A consisted of AccQ Tag Eluent A Concentrate in deionised water (1:10 v/v). The mobile phase B was acetonitrile and the mobile phase C was ultrapure water. The running time was 45 minutes. Gradient elution was described in Supplementary Table S1. Sample injection volume was 5 μL. Flow rate was 1.0 mL min-1. Column temperature was set at 37 °C. Amino acids were detected at 248 nm, and identified by comparison of retention times and spectra of standard solutions of amino acids kit and L-theanine (Joshi et al., 2015; Yu et al., 2014).

2. Is the analytical procedure used for the determnation of FAA validated? Can the authors provide LoD and LoQ? Which are the levels of precision?

Response:

Waters Company has systematically validated the method of determining FAAs by AccQ-Tag-HPLC. This analytical procedure has been widely used in the determination of free amino acids in plants (Dhillon et al., 2014; Joshi et al., 2015; Yu et al., 2014). Previous methodological studies on the determination of FAAs in tea by this method have also been carried out (Zhao et al, 2013; Yu et al., 2014; Xue et al., 2010). Previous data showed that: LoD varied from 0.2 to 19.2 μM, and LoQ varied from 0.7 to 63.9 μM. The RSD of retention time varied from 0.02% to 0.70%, and the RSD of peak areas varied from 0.11% to 5.45% (Zhao et al, 2013; Yu et al., 2014; Xue et al., 2010). Although we did not do a complete methodological validation test, we carried out strict quality control to ensure the precision of our results. We will carry out systematic validation experiments in future research. The corresponding description has been added to the Materials and Methods section as follows (lines 196-200):

For quality control, we inserted the standard samples before, during and after the sample testing. The RSD of the retention time and peak area of each amino acid has been shown in Supplementary Table S2. The RSD of the retention time and peak areas varied from 0.01% to 0.20%, 0.78% to 4.74%, respectively. This suggests that our test conditions are stable.

3. The authors state that "17 types of FAA" were detected. Why only results regarding aspartic acid, glutammic acid and theanine are presented?

Response:

Because the results of principal component analysis showed that aspartic acid, glutammic acid and theanine were the main factors, we focused on the results of these amino acids. The contents of 17 types of FAAs have also been shown in Fig. 2a. The corresponding descriptions have been added to the manuscript as follows (lines 95-98):

These 17 kinds of FAAs included aspartic acid, serine, glutamic acid, glycine, histidine, arginine, threonine, alanine, proline, theanine, cysteine, tyrosine, valine, methionine, lysine, isoleucine and leucine. The content of FAAs varied greatly (varied from 0.02 to 0.38 mg g-1 DW).

4. Can the authors provide a chromatogram of the FAA?

Response:

Thank you for your advice. A chromatogram of the FAAs has been supplied in Supplementary Figure S1. Three graphs in each column represent three biological duplications in one treatment.

5. How many plant samples per group were used for the study?

Response:

Three repeats (each repetition consisted of 20 plants) per group were used for the study. The corresponding description has been added to the Materials and Methods section (lines 177-178).

References

Joshi, R.; Rana, A.; Gulati, A. Studies on quality of orthodox teas made from anthocyanin-rich tea clones growing in Kangra valley, India. Food Chem. 2015, 176, 357-366.

Yu, P.; Yeo, A.S.L.; Low, M.Y.; Zhou, W. Identifying key non-volatile compounds in ready-to-drink green tea and their impact on taste profile. Food Chem. 2014, 155, 9-16.

Dhillon, M.K.; Kumar, S.; Gujar, G.T. A common HPLC-PDA method for amino acid analysis in insects and plants. Indian J. Exp. Biol. 2014, 52, 73-79.

Zhao, M.; Ma, Y.; Dai, L.L.; et al. A High-Performance Liquid Chromatographic Method for Simultaneous Determination of 21 Free Amino Acids in Tea. Food Anal. Method. 2013, 6, 69-75.

Xue, G.; Li, H.Y.; Yu, L.; et al. The Comparison of Aimd Acid in different kinds of Dark Tea. Tea Com. 2010, 37, 9-12 (in Chinese).

Chen, Y.; Han, W.; Tang, L.; Tang, Z.; Fang, J. Leaf nitrogen and phosphorus concentrations of woody plants differ in responses to climate, soil and plant growth form. Ecography2013, 36, 178-184.

He, M.; Ke, Z.; Tan, H.; Rui, H.; Su, J.; Jin, W.; et al. Nutrient levels within leaves, stems, and roots of the xeric species Reaumuria soongorica in relation to geographical, climatic, and soil conditions. Ecol. Evol. 2015, 5, 1494-1503.

Tao, Y.; Ganlin, W.U.; Zhang, Y.; Zhou, X. Leaf N and P stoichiometry of 57 plant species in the Karamori Mountain Ungulate Nature Reserve, Xinjiang, China. J. Arid Land2016, 8, 935-947.

Yang, X.; Huang, Z.; Zhang, K.; Cornelissen, J.H.C.; Nardoto, G.B. C:N:P stoichiometry of Artemisia species and close relatives across northern China: unravelling effects of climate, soil and taxonomy. J. Ecol. 2015, 103, 1020-1031.

Reviewer 2 Report

This is an interesting work, however, the authors should provide some additional data and descriptions (details), particularly in terms of QC/AC of the results achieved.

The analytical results obtained      under different conditions were statistically compared, and statistically      significant differences were marked with lowercase letters “a”, “b”, and      “c”. Unfortunately the authors have not written anything about the test      (and its conditions) that was used by them.

Figure 2a. What is the label of      the x-axis in this figure. If this is a chromatogram (RP-HPLC-DAD), it is      strange that only 3 points are per 1 peak.

l. 168. What were recoveries of      FAAs?

l. 172. More details should be      given about RP-HPLC-DAD conditions used for the determination of up to 17      FAAs.

The authors measured the      content of N and FAAs. In both cases, there is no information and respective      data about the validity of the results achieved. What was the accuracy of      the results obtained with both method and how it was assessed?

Author Response

Dear Editor and reviewer:

Thank you very much for giving us the opportunity to revise our manuscript.

We have studied reviewer’s comments and suggestions carefully and have made revision accordingly which are marked in red in the manuscript. We have also replied point by point to reviewer’s comments in the following letter which are marked in blue. Attached please find the revised version, which we would like to submit for your kind consideration. The English language of our manuscript has been editing by MDPI (english-7369).

We would like to express our great appreciation to you and reviewers for comments on our manuscript. Looking forward to hearing from you.

With best regards!

Yours sincerely,

Liyuan Wang, Hao Cheng

Reviewer 2

1.     This is an interesting work, however, the authors should provide some additional data and descriptions (details), particularly in terms of QC/AC of the results achieved.

Response:

The QC of the results has been shown in Supplementary Table S2. For quality control, we inserted the standard samples before, during and after the sample testing. The RSD of the retention time and peak area of each amino acid has been shown in Supplementary Table S2. The RSD of the retention time and peak areas varied from 0.01% to 0.20%, 0.78% to 4.74%, respectively. This suggests that our test conditions are stable (lines 196-200).

In addition, a chromatogram of the FAAs has been supplied in Supplementary Figure S1. We have described the test conditions in more detail in the Materials and Methods section as follows (lines 182-196):

To determine the FAA content, 0.2 g of dried and ground leaves were extracted by 10 mL pure water at a constant temperature of 100 °C for 30 min. The samples were shaken once in 10 min. Then samples were centrifuged for 10 min at 3500 r min-1. The supernatant was taken and filtered through 0.22 um the membrane filter of water-phase for use. The derivatization of free amino acids was performed using the AccQ-Fluor Reagent Kit according to the manufacturer’s specifications. Briefly, 10 μL of standard amino acid mix solution, and tea extracts were mixed with 70 μL of AccQ Tag borate buffer; then 20 μL of AccQ Tag reagent previously dissolved in 1.0 ml of AccQ Tag reagent diluents were added. The reaction was allowed to proceed for 10 min at 55 °C. Separation was performed on the HPLC system equipped with a Waters AccQ Tag reversed-phase HPLC column (150 mm×3.9 mm, 4 μm). The mobile phase A consisted of AccQ Tag Eluent A Concentrate in deionised water (1:10 v/v). The mobile phase B was acetonitrile and the mobile phase C was ultrapure water. The running time was 45 minutes. Gradient elution was described in Table S1. Sample injection volume was 5 μL. Flow rate was 1.0 mL min-1. Column temperature was set at 37 °C. Amino acids were detected at 248 nm, and identified by comparison of retention times and spectra of standard solutions of amino acids kit and L-theanine [30-31].

2.     The analytical results obtained under different conditions were statistically compared, and statistically significant differences were marked with lowercase letters “a”, “b”, and “c”. Unfortunately the authors have not written anything about the test (and its conditions) that was used by them.

Response:

Thank you for your advice. We use lowercase letters to compare the differences of tea plants under different external nutrient supply conditions (i.e. the split-root system). The different letters showed that means were significantly different at P < 0.05. The different conditions we used are as follows (lines 160-173):

In this split-root system, different N forms were provided in the two separate spaces. At the beginning of the split-root experiment, we selected tea plant seedlings with uniform root and shoot growth. Then, the tea roots were planted uniformly in two separated physical spaces uniformly. Three combinations of different N forms were provided as follows: (1) a homogeneous NH4+ condition (U. NH4+: both compartments had NH4+), (2) a homogeneous NO3- condition (U. NO3-: both compartments had NO3-), and (3) a heterogeneous split condition (Sp. NH4+ / Sp. NO3-: one compartment had NH4+, and the other had NO3-). In the U. NH4+ and Sp. NH4+ treatments, the nutrient solutions contained the following macronutrients (mmol L-1): (NH4)2SO4 (1), KH2PO4 (0.07), K2SO4 (0.3), MgSO4·7H2O (1), CaCl2·2H2O (0.53), and Al2(SO4)3·18H2O (0.035) and micronutrients (μmol L-1) H3BO4 (7), MnSO4·H2O (1), ZnSO4·7H2O (0.67), CuSO4·5H2O (0.13), (NH4)6Mo7O24·4H2O (0.047) and EDTA-Fe (4.2). In the U. NO3- and Sp. NO3- treatments, the nutrient solutions contained the following macronutrients (mmol L-1): Mg(NO3)2 (1), KH2PO4 (0.07), K2SO4 (0.3), CaCl2·2H2O (0.53), and Al2(SO4)3·18H2O (0.035) and the same micronutrients. The pH of the three treatments was 5.0.

In the Figure Caption we added the corresponding sentences (lines 91-93, 118-120, and 153-156): “The mean ± SE (n=3) was shown in the data. The different letters showed that means were significantly different at P < 0.05. The different conditions we used to compare the analytical results obtained were shown in the Materials and Methods section.”

3. Figure 2a. What is the label of the x-axis in this figure. If this is a chromatogram (RP-HPLC-DAD), it is strange that only 3 points are per 1 peak.

Response:

The label of the X-axis is the name of the FAAs, which has been added in Figure 2a. Figure 2a is a figure of the contents of all FAAs in the leaves, rather than a chromatogram (RP-HPLC-DAD). A chromatogram of the FAAs has been supplied in Supplementary Figure S1.

4. 168. What were recoveries of FAAs?

Response:

The recoveries of FAAs varied from 84.6% to 102.3%. This sentence has been added in the Materials and Methods section (lines 200-201).

5. 172. More details should be given about RP-HPLC-DAD conditions used for the determination of up to 17 FAAs.

Response:

More details about the test conditions have been added in the Materials and Methods section as follows (lines 182-196):

0.2 g of dried and ground leaves were extracted by 10 mL pure water at a constant temperature of 100 °C for 30 min. The samples were shaken once in 10 min. Then samples were centrifuged for 10 min at 3500 r min-1. The supernatant was taken and filtered through 0.22 um the membrane filter of water-phase for use. The derivatization of free amino acids was performed using the AccQ-Fluor Reagent Kit according to the manufacturer’s specifications. Briefly, 10 μL of standard amino acid mix solution, and tea extracts were mixed with 70 μL of AccQ Tag borate buffer; then 20 μL of AccQ Tag reagent previously dissolved in 1.0 ml of AccQ Tag reagent diluents were added. The reaction was allowed to proceed for 10 min at 55 °C (Joshi et al., 2015; Yu et al., 2014). Separation was performed on the HPLC system equipped with a Waters AccQ Tag reversed-phase HPLC column (150 mm×3.9 mm, 4 μm). The mobile phase A consisted of AccQ Tag Eluent A Concentrate in deionised water (1:10 v/v). The mobile phase B was acetonitrile and the mobile phase C was ultrapure water. The running time was 45 minutes. Gradient elution was described in Supplementary Table S1. Sample injection volume was 5 μL. Flow rate was 1.0 mL min-1. Column temperature was set at 37 °C. Amino acids were detected at 248 nm, and identified by comparison of retention times and spectra of standard solutions of amino acids kit and L-theanine (Joshi et al., 2015; Yu et al., 2014).

6. The authors measured the content of N and FAAs. In both cases, there is no information and respective data about the validity of the results achieved. What was the accuracy of the results obtained with both method and how it was assessed?

Response:

1)     For measurement of the content of FAAs, Waters Company has systematically validated the method of determining FAAs by AccQ-Tag-HPLC. This analytical procedure has been widely used in the determination of free amino acids in plants (Dhillon et al., 2014; Joshi et al., 2015; Yu et al., 2014). Previous methodological studies on the determination of FAAs in tea by this method have also been carried out (Zhao et al, 2013; Yu et al., 2014; Xue et al., 2010). Previous data showed that: LoD varied from 0.2 to 19.2 μM, and LoQ varied from 0.7 to 63.9 μM. The RSD of retention time varied from 0.02% to 0.70%, and the RSD of peak areas varied from 0.11% to 5.45% (Zhao et al, 2013; Yu et al., 2014; Xue et al., 2010).

In this study, we used the standard samples to evaluate the accuracy of the results. We inserted the standard samples before, during and after the sample testing. The RSD of the retention time and peak area of each amino acid has been shown in Supplementary Table S2. The RSD of the retention time and peak areas varied from 0.01% to 0.20%, 0.78% to 4.74%, respectively. This suggests that our results were accurate.

2)     For measurement of the content of N, Elementar Company has systematically validated the method of determining N by CN element automatic analyzer. This analytical procedure has been widely used in the determination of N in plants (Chen et al., 2013; He et al., 2015; Tao et al., 2016; Yang et al., 2015).

In this study, we used the standard samples to evaluate the accuracy of the results. We inserted the standard samples before, during and after the sample testing. The RSD of the content of N varied from 0.01% to 0.03%. This suggests that our results were accurate.

References

Joshi, R.; Rana, A.; Gulati, A. Studies on quality of orthodox teas made from anthocyanin-rich tea clones growing in Kangra valley, India. Food Chem. 2015, 176, 357-366.

Yu, P.; Yeo, A.S.L.; Low, M.Y.; Zhou, W. Identifying key non-volatile compounds in ready-to-drink green tea and their impact on taste profile. Food Chem. 2014, 155, 9-16.

Dhillon, M.K.; Kumar, S.; Gujar, G.T. A common HPLC-PDA method for amino acid analysis in insects and plants. Indian J. Exp. Biol. 2014, 52, 73-79.

Zhao, M.; Ma, Y.; Dai, L.L.; et al. A High-Performance Liquid Chromatographic Method for Simultaneous Determination of 21 Free Amino Acids in Tea. Food Anal. Method. 2013, 6, 69-75.

Xue, G.; Li, H.Y.; Yu, L.; et al. The Comparison of Aimd Acid in different kinds of Dark Tea. Tea Com. 2010, 37, 9-12 (in Chinese).

Chen, Y.; Han, W.; Tang, L.; Tang, Z.; Fang, J. Leaf nitrogen and phosphorus concentrations of woody plants differ in responses to climate, soil and plant growth form. Ecography2013, 36, 178-184.

He, M.; Ke, Z.; Tan, H.; Rui, H.; Su, J.; Jin, W.; et al. Nutrient levels within leaves, stems, and roots of the xeric species Reaumuria soongorica in relation to geographical, climatic, and soil conditions. Ecol. Evol. 2015, 5, 1494-1503.

Tao, Y.; Ganlin, W.U.; Zhang, Y.; Zhou, X. Leaf N and P stoichiometry of 57 plant species in the Karamori Mountain Ungulate Nature Reserve, Xinjiang, China. J. Arid Land2016, 8, 935-947.

Yang, X.; Huang, Z.; Zhang, K.; Cornelissen, J.H.C.; Nardoto, G.B. C:N:P stoichiometry of Artemisia species and close relatives across northern China: unravelling effects of climate, soil and taxonomy. J. Ecol. 2015, 103, 1020-1031.

Round 2

Reviewer 1 Report

The authors made the required revisions and improved the manuscript. 

Still some minor revisions are needed

Figure S1. Please indicate each peak with a number and report the name of the amino acids in the figure caption

Table S1. Please substitute D% with C%

Please check English in the new added parts

Author Response

Dear reviewer:

Thank you very much for giving us the opportunity to revise our manuscript.

         We have studied reviewer’s comments and suggestions carefully and have made revision accordingly which are marked in red in the manuscript. We have also replied point by point to reviewer’s comments in the following letter which are marked in blue. Attached please find the revised version, which we would like to submit for your kind consideration. The English language of our manuscript has been editing by MDPI (english-7369).

We would like to express our great appreciation to you and reviewers for comments on our manuscript. Looking forward to hearing from you.

With best regards!

Yours sincerely,

Liyuan Wang, Hao Cheng

Reviewer 1

1.      Figure S1. Please indicate each peak with a number and report the name of the amino acids in the figure caption.

Response:

Thank you for your advice. Figure S1 has been changed according to your advice and the corresponding sentence has been added to the figure caption as follows: The numbers on each peak represented the following FAAs: 1-Aspartic acid, 2-Serine, 3-Glutamic acid, 4-Glycine, 5-Histidine, 6-Arginine, 7-Threonine, 8-Alanine, 9-Proline, 10-Theanine, 11-Cysteine, 12-Tyrosine, 13-Valine, 14-Methionine, 15-Lysine, 16-Isoleucine, and 17-Leucine.

2.      Table S1. Please substitute D% with C%.

Response:

Thank you for your advice. D% has been substituted with C% in Table S1.

3.      Please check English in the new added parts.

Response:

The English in the new added parts has been checked and the English language of the whole manuscript has been modified by MDPI (english-7369).

Reviewer 2 Report

The authors have prepared detailed explanations and replies to all previous comments. I am satisfied with these replies and all changes made by the authors in the revised version of the article. These changes certainly have improved the final version of the article.

Author Response

Dear reviewer:

Thank you very much for giving us the opportunity to revise our manuscript.

We have studied reviewer’s comments and suggestions carefully and have made revision accordingly which are marked in red in the manuscript. The English in the new added parts has been checked and the English language of the whole manuscript has been modified by MDPI (english-7369). Attached please find the revised version, which we would like to submit for your kind consideration.

We would like to express our great appreciation to you and reviewers for comments on our manuscript. Looking forward to hearing from you.

With best regards!

Yours sincerely,

Liyuan Wang, Hao Cheng